EMBO
Molecular Medicine

# *In vivo* molecular imaging of chemokine receptor CXCR4 expression in patients with advanced multiple myeloma

Kathrin Philipp-Abbrederis[1,†], Ken Herrmann[2,**,†], Stefan Knop[3], Margret Schottelius[4], Matthias Eiber[5], Katharina Lückerath[2], Elke Pietschmann[1], Stefan Habringer[1,6], Carlos Gerngroß[5], Katharina Franke[1], Martina Rudelius[7], Andreas Schirbel[2], Constantin Lapa[2], Kristina Schwamborn[8], Sabine Steidle[1], Elena Hartmann[7], Andreas Rosenwald[7], Saskia Kropf[9], Ambros J Beer[5,‡], Christian Peschel[1,6], Hermann Einsele[3], Andreas K Buck[2], Markus Schwaiger[5,6], Katharina Götze[1,6], Hans-Jürgen Wester[4,6,9] & Ulrich Keller[1,6,*]

## Abstract

CXCR4 is a G-protein-coupled receptor that mediates recruitment of blood cells toward its ligand SDF-1. In cancer, high CXCR4 expression is frequently associated with tumor dissemination and poor prognosis. We evaluated the novel CXCR4 probe [68Ga]Pentixafor for *in vivo* mapping of CXCR4 expression density in mice xenografted with human CXCR4-positive MM cell lines and patients with advanced MM by means of positron emission tomography (PET). [68Ga]Pentixafor PET provided images with excellent specificity and contrast. In 10 of 14 patients with advanced MM [68Ga]Pentixafor PET/CT scans revealed MM manifestations, whereas only nine of 14 standard [18F]fluorodeoxyglucose PET/CT scans were rated visually positive. Assessment of blood counts and standard CD34+ flow cytometry did not reveal significant blood count changes associated with tracer application. Based on these highly encouraging data on clinical PET imaging of CXCR4 expression in a cohort of MM patients, we conclude that [68Ga]Pentixafor PET opens a broad field for clinical investigations on CXCR4 expression and for CXCR4-directed therapeutic approaches in MM and other diseases.

**Keywords** chemokine receptor; CXCR4; *in vivo* imaging; multiple myeloma; positron emission tomography

**Subject Categories** Biomarkers & Diagnostic Imaging; Cancer

## Introduction

Chemokine receptor-4 (CXCR4) is a member of the G-protein-coupled chemokine receptor family. The sole known natural ligand of CXCR4 is CXCL12/SDF-1. SDF-1 binding to CXCR4 activates downstream signaling pathways such as MAP kinase and the PI3 kinase pathway, ultimately resulting in altered cell adhesion, migration, and homing (Zlotnik *et al*, 2011; Jacobson & Weiss, 2013). CXCR4 is normally expressed on T and B lymphocytes, monocytes, macrophages, neutrophils, and eosinophils, and by hematopoietic stem/progenitor cells (HSPCs) residing within the bone marrow (BM) niche. Antagonizing the CXCR4-mediated retention of HPCs in this niche by means of anti-CXCR4-directed treatment with the CXCR4 antagonist plerixafor allows mobilization of HPCs for autografting upon myeloablative therapies (Brave *et al*, 2010). Plerixafor treatment also mobilizes various lymphocyte populations into the peripheral blood, emphasizing the important role of the SDF-1/CXCR4 axis for lymphocyte trafficking (Kean *et al*, 2011).

1 III. Medical Department of Hematology and Medical Oncology, Technische Universität München, Munich, Germany
2 Department of Nuclear Medicine, Universitätsklinikum Würzburg, Würzburg, Germany
3 Department of Internal Medicine II, Division of Hematology and Medical Oncology, Universitätsklinikum Würzburg, Würzburg, Germany
4 Pharmaceutical Radiochemistry, Technische Universität München, Munich, Germany
5 Department of Nuclear Medicine, Technische Universität München, Munich, Germany
6 German Cancer Consortium (DKTK) and German Cancer Research Center (DKFZ), Heidelberg, Germany
7 Institute of Pathology, Universitätsklinikum Würzburg and CCC Mainfranken, Würzburg, Germany
8 Institute of Pathology, Technischen Universität München, Munich, Germany
9 Scintomics GmbH, Fürstenfeldbruck, Germany
 *Corresponding author. Tel.: +49 89 41407435; fax: +49 89 41404879; E-mail: ulrich.keller@lrz.tum.de
 **Corresponding author. Tel.: +49 931 20135979; fax: +49 931 201635000; E-mail: Herrmann_K1@ukw.de
 †These authors contributed equally to this work
 ‡Present address: Department of Nuclear Medicine, Universität Ulm, Ulm, Germany

Pathological CXCR4 overexpression has been reported in various types of solid cancers and in hematopoietic malignancies such as leukemia and lymphoproliferative malignancies (Burger & Peled, 2009; Cojoc *et al*, 2013). In cancer, CXCR4 overexpression and receptor activation by SDF-1 binding are key triggers for tumor growth and progression, invasiveness, and metastasis (Muller *et al*, 2001). Accordingly, CXCR4 overexpression has been identified as an adverse prognostic factor in various malignancies (Spano *et al*, 2004; Spoo *et al*, 2007). In particular, CXCR4-mediated interaction that holds cancer (re-)initiating cells within a protective tumor microenvironment (TME) seems to be responsible for resistance to pharmacological treatment, and for relapse, at least in hematopoietic cancers (Teicher & Fricker, 2010; Mendelson & Frenette, 2014; Shain & Tao, 2014).

Multiple myeloma (MM, plasma cell myeloma) is the second most prevalent B-cell cancer. Despite the availability of potent novel drugs, it remains, for the large part of patients, an incurable disease (Palumbo & Anderson, 2011; Ocio *et al*, 2014). MM is characterized by the expansion of malignant plasma cells predominantly within the BM. One key clinical characteristic is the uncoupling of bone formation and bone destruction, resulting in osteolytic bone lesions (Raab *et al*, 2009; Roodman, 2010). Studies on both cultured and patients' primary MM cells showed a strong correlation between CXCR4/SDF-1 activation and MM-related bone disease (Zannettino *et al*, 2005; Bao *et al*, 2013). SDF-1 engages CXCR4 on MM cells favoring their recruitment to the BM by affecting migration, adhesion, and extravasation (Parmo-Cabanas *et al*, 2004; Aggarwal *et al*, 2006; Alsayed *et al*, 2007). Besides the different cell types constituting the BM niche, primary MM cells themselves secrete SDF-1, which results in autocrine stimulation of plasma cell proliferation. Therefore, the CXCR4/SDF-1 axis represents a highly relevant molecular target in MM and other cancers due to its important role in pathogenesis and its potential involvement as a mediator of resistance to treatment (Burger & Kipps, 2006).

Despite the fundamental role of CXCR4 in cancer and in particular MM biology and its significance as a target for therapeutic approaches, a highly sensitive method for CXCR4 assessment and quantification *in vivo* has been lacking so far. Such *in vivo* assessment of CXCR4 expression could provide an additional and clinically important method to select patients for CXCR4-directed treatment, for example, by anti-CXCR4 antibodies that are in early-phase clinical trials (Kashyap *et al*, 2012) (e.g., ClinicalTrials.gov identifier NCT01359657), or for use within a theranostic peptide receptor radiotherapy (PRRT) concept. To meet this clinical need, [$^{68}$Ga]Pentixafor ([$^{68}$Ga]CPCR4.2), a high-affinity CXCR4-targeted nuclear probe for PET imaging, has recently been developed (Demmer *et al*, 2011; Gourni *et al*, 2011) and evaluated in a first proof-of-concept investigation (Wester *et al*, 2015). The present study aimed at the evaluation of [$^{68}$Ga]Pentixafor PET/CT as a novel and powerful tool for sensitive, non-invasive *in vivo* quantification of CXCR4 in preclinical models of MM and in a clinical pilot assessment investigating patients with advanced MM.

# Results

### Frequency of MM patients with high tumor CXCR4 (CD184) expression

The CXCR4-SDF-1 axis constitutes a central mechanism for recruiting and retaining MM cells within the TME (Zannettino *et al*, 2005;

Bao *et al*, 2013). To investigate the frequency of MM patients with high CXCR4 expression, we performed flow cytometry on an unselected cohort of 25 patients undergoing BM biopsy for previously established MM, or because of newly diagnosed monoclonal gammopathy. By assessing CXCR4 expression as compared to an isotype control antibody (gating strategy depicted in Supplementary Fig S1), we identified 14 of 25 patients (56%) with CXCR4-positive MM. Representative histograms depicting CXCR4$^+$ vs. CXCR4$^−$ MM are shown in Fig 1A.

In order to estimate the magnitude of CXCR4 expression in MM cells compared to normal non-malignant cell populations that have previously been described as CXCR4 positive (Aiuti *et al*, 1999; Honczarenko *et al*, 1999; Burger & Kipps, 2006; Brave *et al*, 2010), we compared CXCR4 expression levels in individual MM patient samples judged CXCR4 positive by flow cytometric and immunohistochemical assessment (CXCR4$^+$ MM). In CXCR4$^+$ MM, relative plasma cell surface CXCR4 expression levels were significantly higher than those on intraindividual CD19$^+$ B cells, CD3$^+$ T cells, CD34$^+$ HSPCs, and CD14$^+$ monocytes (Fig 1B–D; gating strategy depicted in Supplementary Fig S1; representative data shown in Supplementary Fig S2A–D), indicating that a CXCR4-directed PET tracer could be suitable for MM imaging.

Thus, in our unselected cohort, over 50% of MM patients expressed CXCR4 on their plasma cells. Relative cancer cell CXCR4 expression in these patients was high compared to intraindividual control BM cell populations.

### [$^{68}$Ga]Pentixafor is a PET tracer suitable for detecting CXCR4$^+$ MM *in vivo*

Considering the high CXCR4 expression levels in a substantial proportion of MM patients as compared to intraindividual control cell populations, we searched for MM cell lines that could be suited for preclinical *in vivo* imaging studies. Considerable levels of *CXCR4* transcript (Fig 2A) and protein (Fig 2B) were detected in the well-established MM lines MM.1S and OPM-2 as opposed to the ovarian cancer cell line HeLA, which is characterized by low CXCR4 expression. Moreover, MM.1S and OPM-2 cells were found to bind the CXCR4-directed PET probe [$^{68}$Ga]Pentixafor (Fig 2C). Thus, these cell lines represent models for *in vivo* binding and uptake studies.

In order to determine the suitability of the high-affinity human CXCR4-specific probe Pentixafor as an *in vivo* MM PET tracer, NOD SCID mice were xenografted with MM.1S and OPM-2 cells and underwent consecutive [$^{18}$F]FDG and [$^{68}$Ga]Pentixafor PET. Imaging with [$^{68}$Ga]Pentixafor resulted in markedly higher mean tumor-to-background ratios (TBR) for both the MM.1S and OPM-2 xenografted tumors, compared to the widely used tracer [$^{18}$F]FDG (Fig 2D and E). Flow cytometric quantification of cell surface CXCR4 expression on resected tumors suggested a correlation between CXCR4 cell surface levels and the [$^{68}$Ga]Pentixafor uptake in the respective xenografts observed in the small animal PET studies (Fig 2F and G). As expected when using a human CXCR4-specific probe such as Pentixafor (Gourni *et al*, 2011), virtually no tracer uptake was observed in CXCR4-expressing mouse organs such as spleen, lung, adrenals, or the BM (Fig 2E). To further substantiate the specificity of [$^{68}$Ga]Pentixafor binding to human

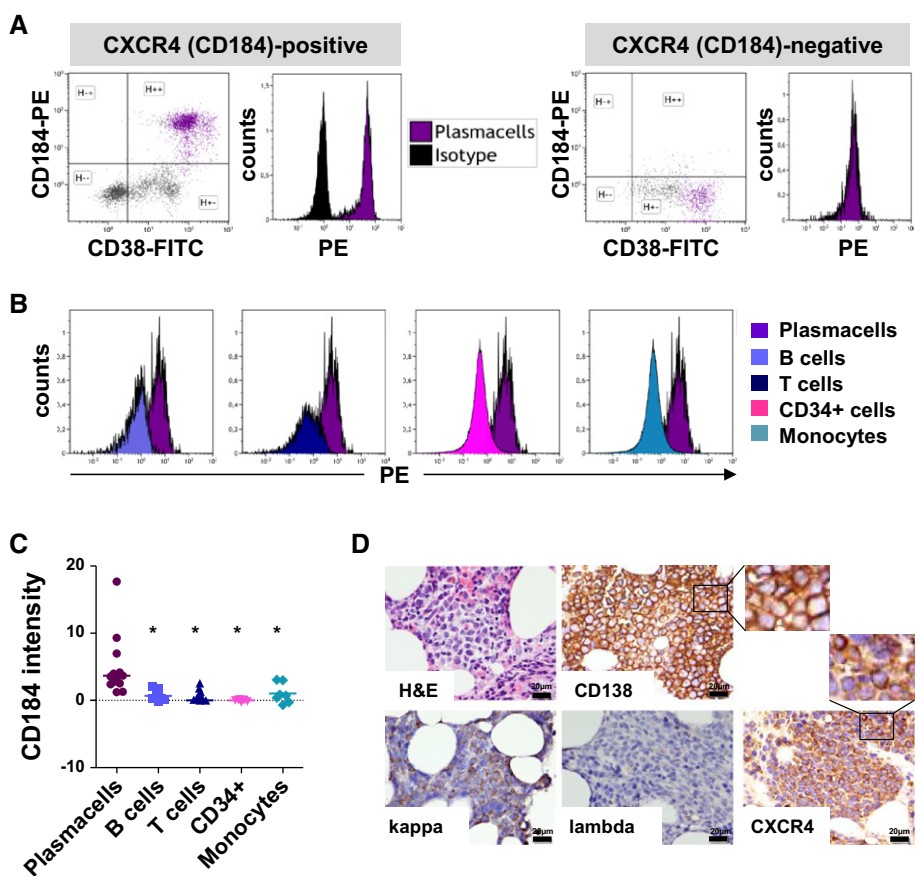

**Figure 1. CXCR4 expression in MM patient bone marrow.**

A   Flow cytometric evaluation of CXCR4 surface expression using an anti-CXCR4-PE antibody. Left: positive patient; right: negative patient; representative data are shown. The gating strategy is depicted in Supplementary Fig S1.

B   Representative histograms revealing the magnitude of CXCR4 expression in myeloma cells as compared to the indicated bone marrow cell subtype.

C   Median fluorescence intensity of surface CXCR4 expression relative to isotype control ($n = 7$–$10$ patients). Horizontal bars indicate mean of all individual patient values; asterisks indicate statistical significance (plasma cells vs B cells: $P = 0.0082$; plasma cells vs T cells: $P = 0.0091$; plasma cells vs CD34$^+$ cells: $P = 0.011$; plasma cells vs monocytes: $P = 0.0423$; Student's $t$-test to compare mean plasma cell relative expression with mean relative expression of each further indicated cell subtype). MM patients judged positive for CXCR4 expression were selected for this analysis.

D   Representative MM bone marrow staining of a patient positive for MM CXCR4 expression: hematoxylin and eosin (H&E) staining. Immunohistochemistry for CD138, CXCR4, light chain kappa, and light chain lambda.

CXCR4, we next performed competition studies where mice bearing MM xenograft tumors received the FDA-approved drug AMD3100 (Plerixafor) (Brave *et al*, 2010) before receiving the [68Ga]-labeled PET probe. AMD3100 pretreatment resulted in a near complete loss of [68Ga]Pentixafor binding *in vitro* and *in vivo* (Fig 2H, Supplementary Fig S3).

Thus, [68Ga]Pentixafor is a PET tracer that binds human CXCR4 expressed on MM cell lines and xenograft tumors with high specificity and is suitable as an *in vivo* CXCR4 PET imaging probe.

## [68Ga]Pentixafor provides additional diagnostic information to [18F]FDG in MM patients

To evaluate the suitability of [68Ga]Pentixafor for *in vivo* imaging of MM in patients and for its usefulness to select patients for future CXCR4-directed treatments, we visually analyzed 14

patients with histologically proven, advanced MM. The patient characteristics are shown in Table 1. All patients gave written informed consent for receiving the [68Ga]Pentixafor PET as well as undergoing a standard [18F]FDG PET. Representative images of one [68Ga]Pentixafor PET-positive patient are shown in Fig 3A–D. Representative images of one [68Ga]Pentixafor PET-negative patient are shown in Supplementary Fig S4. In summary, 9 of 14 (64%) [18F]FDG scans were rated visually positive, whereas 10 of 14 (71%) [68Ga]Pentixafor scans revealed disease manifestations (Fig 4A). Visual comparison of [18F]FDG and [68Ga]Pentixafor scans resulted in comparable findings in 3 (21%) patients. In 7 patients (50%), the [68Ga]Pentixafor signal was superior to [18F]FDG identifying more tumor lesions, whereas in 2 patients (14%), [18F]FDG provided additional information compared to [68Ga]Pentixafor. In the remaining 2 patients (14%), [68Ga]Pentixafor and [18F]FDG provided complementary information regarding the detection of myeloma manifestations (Fig 4B). More than three

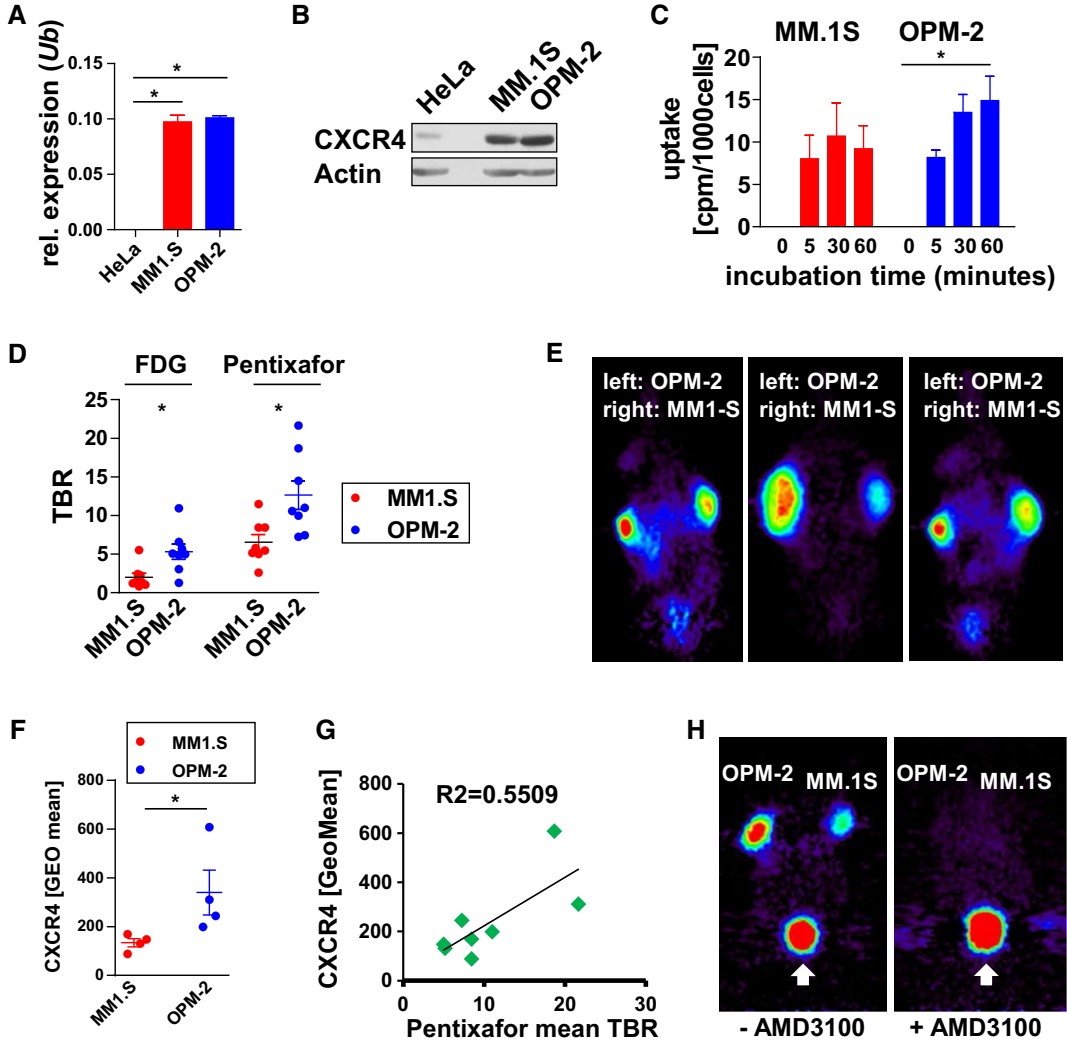

**Figure 2.** [<sup>68</sup>Ga]Pentixafor PET imaging of MM xenografts.

A   Real-time PCR analysis of *cxcr4* transcript expression levels in HeLa (negative control) and in MM cell lines MM.1S and OPM-2. Shown is the mean relative expression ± SEM (*n* = 3 independent experiments). Values are normalized to the expression of ubiquitin (*Ub*). The asterisk indicates statistically significant differences (HeLa vs MM.1S: *P* < 0.001; HeLa vs OPM-2: *P* < 0.0001; Student's *t*-test).

B   CXCR4 protein expression assessed by immunoblotting (one representative blot out of 3 is shown).

C   Binding of [<sup>68</sup>Ga]Pentixafor to MM.1S and OPM-2 cells after the indicated incubation periods (*n* = 3 per cell line and time point). Shown is the mean ± SEM. The difference between the OPM-2 groups is statistically significant; *\*P* < 0.0017 (one-way ANOVA).

D   Mean tumor-to-background ratio (TBR) for [<sup>18</sup>F]FDG (left) and for [<sup>68</sup>Ga]Pentixafor (right) in MM.1S and OPM-2 xenograft-bearing NOD SCID mice. Shown is the mean ± SEM, *n* = 8 tumors (4 mice); *\*P* = 0.0111 for [<sup>18</sup>F]FDG and *\*P* = 0.0113 for [<sup>68</sup>Ga]Pentixafor (Student's *t*-test). One-way ANOVA revealed significant differences between the groups; *P* < 0.0001 (not graphically shown).

E   Representative [<sup>68</sup>Ga]Pentixafor PET images of three mice bearing MM.1S (right shoulder) and OPM-2 (left shoulder) tumors.

F   Flow cytometric quantification of CXCR4 cell surface expression on resected MM.1S and OPM-2 tumors. Data are the mean ± SEM, *n* = 4. *\*P* = 0.0286; Mann–Whitney *U*-test.

G   Correlation of [<sup>68</sup>Ga]Pentixafor PET mean TBR and CXCR4 cell surface expression assessed by flow cytometry. *n* = 8 tumors were analyzed.

H   Mice (*n* = 4) bearing OPM-2 and MM.1S xenografts were coinjected with AMD3100 (right image, one representative mouse) or not pretreated (left image) before undergoing [<sup>68</sup>Ga]Pentixafor PET. The white arrows point to the bladder. Quantification is shown in Supplementary Fig S3A.

Source data are available online for this figure.

lesions were reported in 8 of 14 FDG scans and 8 of 14 [<sup>68</sup>Ga]Pentixafor PET scans. Extramedullary disease (EMD) was detected in 3 [<sup>68</sup>Ga]Pentixafor scans and in 2 [<sup>18</sup>F]FDG PET scans. In one patient, [<sup>68</sup>Ga]Pentixafor but not [<sup>18</sup>F]FDG identified EMD. In comparison with the PET scans, only 1 of 14 CT scans did not show MM manifestations resulting in 13 of 14 (93%) positive scans. More than three lesions were described in 10 of 14 patients, whereas EMD was only reported in 1 patient. An exemplary patient where [<sup>68</sup>Ga]Pentixafor imaging provided superior information is shown in Supplementary Fig S5.

In summary, combined Pentixafor/FDG PET imaging provides additional information on disease extent in MM patients.

**Table 1.  Patient characteristics at initial diagnosis (imaging cohort).**

| | No of patients | % | Median (range) |
|---|---|---|---|
| | 14 | 100 | |
| Age | | | 72 (51–84) |
| Sex | | | |
| male | 11 | 79 | |
| female | 3 | 21 | |
| Monoclonal protein | | | |
| IgG | 6 | 43 | |
| IgA | 2 | 14 | |
| Light chain kappa | 3 | 21 | |
| Light chain lambda | 3 | 21 | |
| Stage of disease at first diagnosis | | | |
| IA | 2 | 14 | |
| IIA | 1 | 7 | |
| IIIA | 5 | 36 | |
| IIIB | 6 | 43 | |
| No of previous regimens | | | 4 (1–9) |
| High-dose chemotherapy | 12 | 86 | |
| Radiotherapy | 6 | 43 | |

## Lesion-based visual and semi-quantitative comparison of [18F]FDG and [68Ga]Pentixafor uptake

Up to three lesions per patient were semi-quantitatively evaluated accounting for a total of 32 lesions (3 lesions in ten patients and 2 lesions in one patient). For [18F]FDG, a total of 23 lesions were rated visually positive as opposed to 26 for [68Ga]Pentixafor. While 17 lesions were read as [18F]FDG and [68Ga]Pentixafor positive, 6 lesions only showed increased [18F]FDG uptake. In contrast, 9 lesions were [68Ga]Pentixafor positive and [18F]FDG negative. The corresponding mean $SUV_{max}$ value for [18F]FDG was 5.5 (range 2.3–15.7) and thus significantly lower than the mean $SUV_{max}$ for [68Ga]Pentixafor (8.7, range 1.4–33.7; $P = 0.018$; Wilcoxon signed-rank test). $SUV_{max}$ data are summarized in Supplementary Fig S6.

## Correlation with standard imaging techniques CT and magnetic resonance imaging (MRI)

Low-dose CT information was available in all 14 patients revealing bone involvement in all but one patient (93%). A diagnostic CT with contrast media was only available in 2 patients, and a diagnostic CT without contrast media in additional 3 patients. With this limitation in mind, EMD was only seen in 1 patient. MRI was performed as part of PET/MRI in two patients showing extended bone disease in both patients with no EMD. The sensitivity of the [68Ga]Pentixafor probe for detecting BM infiltration that is not clearly evident as lytic bone lesions is exemplified in a CXCR4-PET-positive patient, where the [68Ga]Pentixafor PET showed visual positivity corresponding well to intramedullary MM infiltration revealed by MRI (Fig 5A–C).

## [68Ga]Pentixafor PET is not associated with HSPC mobilization or blood count variations

Application of [68Ga]Pentixafor was well tolerated. Peripheral blood was obtained from 7 of 14 patients of the imaging cohort 1 h before (− 1), 1 h (+ 1), 24 h (+ 24), and 7 days (day 7) after [68Ga]Pentixafor PET imaging. White blood counts, hemoglobin, and platelet counts were assessed at the indicated time points and did not reveal significant intrapatient changes associated with tracer application. We also did not observe significant changes in peripheral blood CD34$^+$

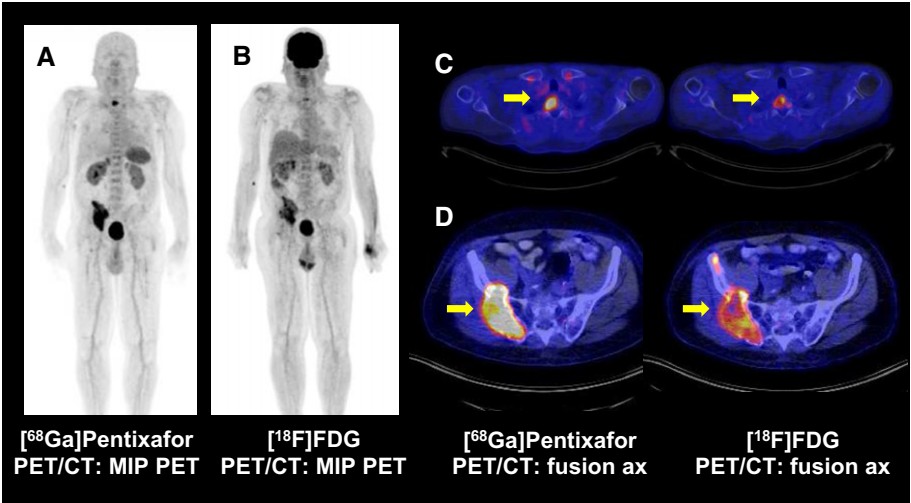

**Figure 3.   [68Ga]Pentixafor PET/CT and [18F]FDG PET/CT.**

A–D   Maximum intensity projections (MIP) of [68Ga]Pentixafor (A) and [18F]FDG PET/CT (B) of a 68-year-old male with histologically proven multiple myeloma indicating the better lesion-to-background contrast for [68Ga]Pentixafor in the corresponding myeloma manifestations. Trans-axial views of the upper thorax (C) and the pelvis (D) underline the higher uptake values of the bone manifestations (yellow arrows) of [68Ga]Pentixafor compared to [18F]FDG.

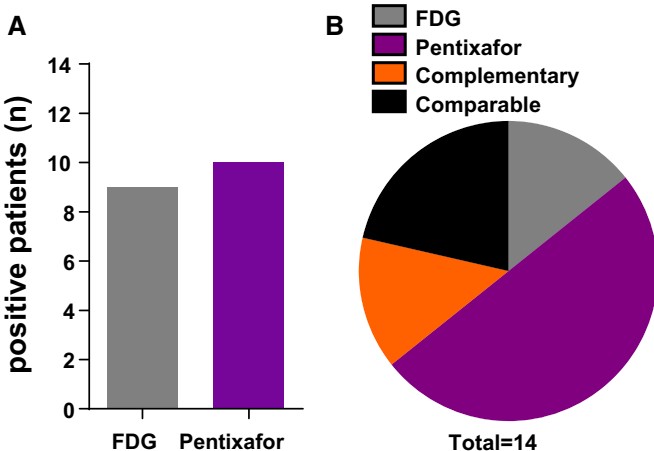

**Figure 4. Visual comparison of [¹⁸F]FDG- and [⁶⁸Ga]Pentixafor PET scans.**

A  Number of patients with visual positivity for the indicated PET tracer (total: $n = 14$).

B  Number of patients (total $n = 14$) for whom imaging with [¹⁸F]FDG PET (FDG, $n = 2$) or [⁶⁸Ga]Pentixafor PET (Pentixafor, $n = 7$) was superior, with comparable positivity (comparable, $n = 3$), and with dual imaging providing complementary visual information (complementary, $n = 2$).

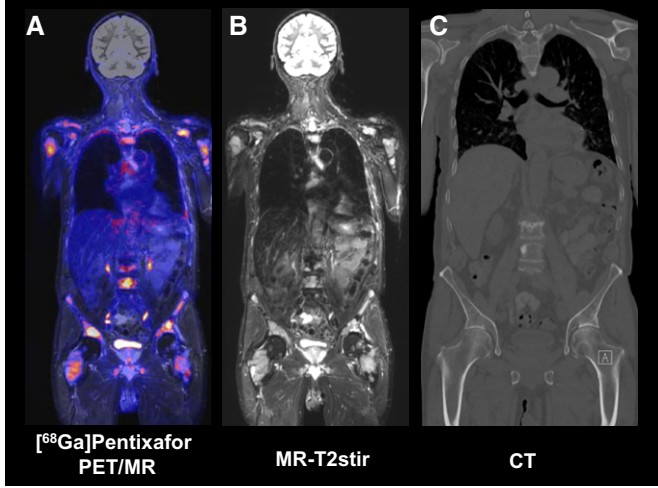

**Figure 5. [⁶⁸Ga]Pentixafor PET/MR images.**

A–C  Coronal views of [⁶⁸Ga]Pentixafor (A), T2 STIR weighted MRI (B) and CT bone window (C) of a male patient with histologically proven multiple myeloma. The increased [⁶⁸Ga]Pentixafor uptake correlates with the hyperintense T2 STIR signal; however, the myeloma manifestations are underestimated in the corresponding bone window CT.

proportion nor absolute number (Supplementary Fig S7). No further toxicities were observed during or after [⁶⁸Ga]Pentixafor application.

## Discussion

Our data represent the first study on quantitative PET imaging of CXCR4, a key chemokine receptor involved in leukocyte attraction,

hematopoietic stem cell homing, tumorigenesis, and many other processes, in preclinical models of myeloma and in a cohort of patients with advanced MM.

High CXCR4 expression has been reported in numerous solid cancers and in various hematopoietic malignancies, including MM (Teicher & Fricker, 2010; Weilbaecher et al, 2011). Importantly, the level of CXCR4 expression assessed by either transcript or whole-cell protein-level analysis is not necessarily representative of CXCR4 expression level on the cell surface (Teicher & Fricker, 2010). CXCR4 overexpression on the cell membrane is, however, the key parameter for successful CXCR4-directed tumor targeting *in vivo*, both for diagnostic imaging and in particular for endoradiotherapeutic approaches. Our in-depth analysis of intraindividual surface CXCR4 levels in BM subpopulations revealed that in at least half of the patients, CXCR4 expression on the surface of MM cells is significantly elevated as compared to B and T cells, monocytes, and CD34⁺ HSPCs. Negligible levels of CXCR4-specific binding were also observed in normal BM and the main lymphatic organs spleen, thymus, and lymph nodes. Thus, given the high contrast obtained in [⁶⁸Ga]Pentixafor PET between low endogenous CXCR4 expression and pathological CXCR4 overexpression of positive MM lesions, this imaging methodology may allow for selecting patients for CXCR4-directed treatment, such as radionuclide or toxin-labeled Pentixafor analogs. In accordance with the frequent and strong presence of CXCR4 on plasma cells from MM patients, CXCR4 expression on transcriptional and protein level in established human MM cell lines was robust and [⁶⁸Ga]Pentixafor was found to bind to these cells *in vitro*. Additionally, μPET imaging with Pentixafor detected xenografted MM tumors with high specificity and contrast, and tracer accumulation was found to correlate with CXCR4 cell surface expression in tumor tissue. Thus, detection and investigation of MM lesions *in vivo* seems to be feasible with [⁶⁸Ga]Pentixafor. However, the high tumor/background ratios observed in the mouse xenograft model are at least partly the result of the selectivity of [⁶⁸Ga]Pentixafor for human CXCR4 (Demmer et al, 2011). Since [⁶⁸Ga]Pentixafor does not bind to murine CXCR4, [⁶⁸Ga]Pentixafor PET studies investigating CXCR4-associated pathologies in a mouse model are currently limited to xenograft animal models.

High levels of CXCR4 have been shown to indicate particularly aggressive disease, metastasis, or poor prognosis in solid cancers (Teicher & Fricker, 2010; Weilbaecher et al, 2011) and AML (Spoo et al, 2007). Such correlations are, however, not expected in MM, a malignancy that is at primary diagnosis nearly exclusively BM based (Raab et al, 2009; Roodman, 2010; Palumbo & Anderson, 2011; Ocio et al, 2014). Although CXCR4/SDF-1 activation and MM-related bone disease are clearly associated (Zannettino et al, 2005; Bao et al, 2013), it is evident that SDF-1 engages CXCR4 on MM cells favoring their recruitment to the BM by affecting migration, adhesion, and extravasation (Parmo-Cabanas et al, 2004; Aggarwal et al, 2006; Alsayed et al, 2007). Thus, losing the requirement for MM–TME interactions in situations like plasma cell leukemia or in extramedullary relapse could point to a scenario where [⁶⁸Ga]Pentixafor PET might be inferior to standard [¹⁸F]FDG PET, which has proven diagnostic value in extramedullary MM (Bao et al, 2013; Stessman et al, 2013). Although we investigated such patients with EMD after allogeneic SCT with both PET tracers, it is to date not clear whether this is the case. Our data, however, suggest a possible complementary benefit when both PET tracers

are available. For example, [68Ga]Pentixafor did not penetrate the intact blood–brain barrier, in sharp contrast to FDG.

The CXCR4 antagonist Plerixafor is an FDA- and EMA-approved drug used for mobilization of HSPCs for stem cell graft retrieval (Brave *et al*, 2010). The dosage used for this purpose is 0.24 mg/kg body weight per day, in combination with granulocyte colony-stimulating factor (G-CSF), and usually applied after several days of G-CSF treatment alone. As expected on the basis of the applied amount of Pentixafor (< 20 μg/patient), we did not observe any side effects of [68Ga]Pentixafor application acutely or during the course of several days, in particular any blood count or HSPC abnormalities other than caused by the underlying MM. Also, toxicity of the radionuclide 68Ga in the administered dose is negligible. Previous efforts to use Plerixafor-derived probes for imaging purposes failed due to an accumulation of the drug in spleen tissue, or by unspecific binding within liver tissue or liver-based metabolism (Nimmagadda *et al*, 2009, 2010; De Silva *et al*, 2011; Kuil *et al*, 2012; Weiss & Jacobson, 2013). In none of our patients did we find evidence of increased binding of [68Ga]Pentixafor in the liver that was not associated with MM as documented with the established PET tracer [18F]FDG.

In the vast majority of cases, MM presents as a systemic disease with a measurable monoclonal gammopathy that allows following response to treatment and observing the patient for relapse non-invasively (Rajkumar *et al*, 2011). Analysis of serum and urine by means of immunofixation allows for the detection of subclinical disease, and low-dose whole-body computed tomography is a powerful tool for the detection of osteolytic lesions which may prompt initiation of radiotherapy planning (Ippolito *et al*, 2013).

Magnetic resonance imaging (MRI) provides more detailed information about BM infiltration and EMD in MM (Baur-Melnyk *et al*, 2005), and [18F]FDG PET imaging clearly provides additional information with regard to prognosis and extent of local disease, in particular EMD (Durie *et al*, 2002; Zamagni *et al*, 2011; Agarwal *et al*, 2013). Both MRI and [18F]FDG PET are, however, not considered a routine procedure required for every patient (Rajkumar *et al*, 2011). The purpose of this study was thus not to establish a novel diagnostic PET tracer for MM in particular, but to test the applicability of this molecular probe in a disease with frequent CXCR4 surface expression. More importantly, however, we consider MM a disease where it would be highly interesting to combine [68Ga]Pentixafor PET as a selection marker for CXCR4-directed treatment and to ensure target expression. Upon availability, [177Lu]- or [90Y]-coupled Pentixafor analogs could become attractive radiopharmaceuticals for a theranostic approach with 68Ga-labeled Pentixafor as a marker for patient selection and therapy monitoring, and the latter compounds as endoradiotherapeutics. Such an approach has previously been demonstrated in neuroendocrine tumors with resounding success (Breeman *et al*, 2003; Das *et al*, 2007; Werner *et al*, 2014). The most obvious and attractive scenario would thus follow the sequence [68Ga]Pentixafor PET, therapy with a radionuclide or toxin, a standard myeloablative therapy, followed by SCT. SCT thus would allow applying labeled Pentixafor doses that will most likely result in perturbations of BM function. Such an approach is currently under investigation. Also, disintegrating and targeting the HSC interaction with its BM niche could be an interesting principle in treating HSPC disorders (Burger *et al*, 2009; Shain & Tao, 2014). In addition to such direct usage of Pentixafor as a carrier for active agents, the use as an

imaging modality for patient selection would be an obvious approach, for example, for anti-Notch-directed treatments that are currently being evaluated and involve CXCR4-SDF-1 activation (Mirandola *et al*, 2013), or for selection of patients receiving anti-CXCR4 therapies such as BMS-936564/MDX-1338, a fully human anti-CXCR4 antibody currently in clinical investigation (Kuhne *et al*, 2013).

Our data demonstrate the suitability of [68Ga]Pentixafor for PET imaging of CXCR4 chemokine receptor expression in MM patients. We conclude that this novel PET tracer could serve as an innovative imaging agent, for *in vivo* biomarker identification that could result in patient selection for CXCR4-directed treatments, and eventually for receptor-radio(drug)peptide therapy.

# Materials and Methods

### Patients

Detailed characteristics for the PET imaging patient cohort are given in Table 1. All patients had histologically proven MM and active ongoing disease as assessed by biopsy or immune electrophoresis. As previously reported for other [68Ga]-labeled peptides (Haug *et al*, 2014), [68Ga]Pentixafor was administered under the conditions of pharmaceutical law (The German Medicinal Products Act, AMG §13 2b) according to the German law and in accordance with the responsible regulatory agencies (Regierung von Oberbayern, Regierung von Unterfranken). All patients gave written informed consent prior to the investigations. The responsible ethics committees of the Technische Universität München and the Universitätsklinikum Würzburg approved data analysis.

The current study is not a confirmatory one. There were no prespecified hypotheses that would have allowed for sample size calculation. It is an observational pilot study used to conduct explorative analyses. Therefore, the sample size was chosen to serve this purpose. It enabled the computation of descriptive and explorative statistics.

### Cell lines and cell culture

The human multiple myeloma cell lines OPM-2 (DSMZ no. ACC50) and MM.1S (ATCC CRL-2974) were cultured in RPMI 1640 supplemented with 10% FCS, 2 mM L-glutamine, 1 mM sodium pyruvate, 100 U/ml penicillin, and 100 μg/ml streptomycin. HeLa cells were cultured in DMEM supplemented with 10% FCS, 100 U/ml penicillin, and 100 μg/ml streptomycin. Cells were maintained at 37°C in a 5% $CO_2$ humidified atmosphere. All media and supplements were obtained from Invitrogen (Darmstadt, Germany).

### Mice and tumor xenograft experiments

Animal studies were performed in agreement with the Guide for Care and Use of Laboratory Animals published by the US National Institutes of Health (NIH Publication No. 85-23, revised 1996), in compliance with the German law on the protection of animals, and with approval of the responsible regional authorities (Regierung von Unterfranken). NOD.CB17-*Prkdc^scid*/NCrHsd mice were bred at the animal facility at the Center of Experimental and Molecular Medicine (ZEMM) of the University of Würzburg. Equally housed and fed female mice of same size and age were randomly distributed

into experimental groups. A total of $5 \times 10^6$ MM.1S or OPM-2 cells in 100 μl PBS were injected subcutaneously into the shoulder region of approximately 8-week-old animals. Tumor growth was monitored using a shifting calliper. Imaging experiments were initiated when tumor size reached 200–300 mm$^3$. Mouse experiment samples sizes were chosen to allow descriptive and explorative statistical analysis.

### RNA and protein extraction, real-time PCR and immunoblotting

RNA extraction was performed using the RNeasy Mini Kit (Qiagen, Hilden, Germany). cDNA synthesis was performed using the Omni-script RT Kit according to the manufacturer's protocol (Qiagen, Hilden, Germany). Real-time PCR was performed using Platinum SYBR-Green qPCR SuperMix-UDG (Invitrogen) on an ABI Prism 7700 (Applied Biosystems). Data analysis was performed by comparing Ct values with a control sample set as 1. Sequences for primers are available upon request. Protein extracts (30 μg per lane) were electrophoretically separated on a SDS–PAGE gel, transferred to membranes (Millipore, Darmstadt, Germany), and blotted with antibodies specific for CXCR4 (clone UMB2; Abcam, Cambridge, UK) and β-actin (clone AC-74; Sigma-Aldrich, Taufkirchen, Germany).

### Flow cytometry

White blood cell counts, hemoglobin and platelet counts were measured using an Advia 120 (Siemens, Erlangen, Germany).

Blood samples from the patients of the imaging cohort were collected in heparin and filtered. Circulating CD34$^+$ cell counts were assessed in peripheral blood by standardized and certified single-platform flow cytometry on a Cytomics FC 500 analyzer (Beckman Coulter, Krefeld, Germany) at four different time points. Cells were stained with antibodies to CD34 and CD45 (Stem-Kit IM 3630, clone 581; CD45-ECD, clone J33; Beckman Coulter).

Bone marrow samples from an unselected MM patient cohort were collected in heparin tubes and filtered. These analyses were performed upon signed informed consent of all patients to analysis of samples for scientific purposes in an anonymized fashion only. Cell populations of interest were selected by sequential gating using Kaluza Flow Analysis Software (Beckman Coulter). The gating strategy is based on CD45 staining versus side-angle light scatter (SSC) properties as a primary gate to separate CD45-positive lymphocytes, monocytes, and CD45low progenitors from CD45-negative plasma cells. Plasma cells were then identified by CD45/CD38 gating and CD19 negativity. Colored back-gating was used to ensure correct gating of all subpopulations. For statistical data analysis, intensity of the isotype control on each subpopulation was subtracted from the median fluorescence intensity of surface CXCR4 expression in that population. Details on the gating strategy used are given in Supplementary Figs S1 and S2. The following antibodies were used: Beckman Coulter: CD20-ECD (clone B9E9), CD45-ECD (clone J33), CD3-PC5 (clone UCHT1), CD138-PC5 (clone BB4), CD14-PC5 (clone RM052), CD34-FITC (clone 581), CD38-FITC (clone T16), CD19-PC7 (clone J4.119), CD56-PC7 (clone N901), and CD33-PC7 (clone D3HL60.251); BD Pharmingen: CXCR4-PE (clone 12G5).

Surface CXCR4 levels of xenografted human MM cell lines were determined by flow cytometry (BD FACSCalibur, Beckton-Dickinson, Heidelberg, Germany) using an anti-CXCR4-PE antibody (hCD184;

clone 12G5; Miltenyi, Bergisch-Gladbach, Germany) according to the manufacturer's instructions. Data were analyzed using CellQuest software (Beckton-Dickinson).

### Synthesis of [68Ga]Pentixafor

Synthesis of [68Ga]Pentixafor was performed at both centers in a fully automated, GMP-compliant procedure using a GRP module (SCINTOMICS GmbH, Germany) equipped with disposable single-use cassette kits (ABX, Germany), using the method (Demmer, Gourni *et al*, 2011 and Gourni, Demmer *et al*, 2011) and standardized labeling sequence previously described (Martin *et al*, 2014). Prior to injection, the quality of [68Ga]Pentixafor was assessed according to the standards described in the European Pharmacopoeia for [68Ga]-Edotreotide (European Pharmacopoeia; Monograph 01/2013:2482; available at www.edqm.eu).

### *In vitro* binding study

Binding of [68Ga]Pentixafor to MM.1S and OPM-2 MM cells was investigated using a modified standard protocol (Lückerath *et al*, 2013). Briefly, $4 \times 10^5$ cells in 500 μl PBS were incubated with $1 \times 10^6$ counts per minute (cpm) radiotracer/50 μl PBS (equaling approximately 1 nM peptide per sample) for the indicated times. After the removal of unbound tracer and washing with PBS, binding of [68Ga]Pentixafor was quantified using a gamma-counter (Wallac1480-Wizard, Perkin-Elmer, Rodgau, Germany). All samples were measured in triplicate and corrected for background activity and decay. For competition experiments, MM cell lines were pretreated for 30 min with AMD3100 100 μM (Selleck Chemicals, Houston, TX, USA, ordered from Absource Diagnostics GmbH, Munich, Germany) before being objected to [68Ga]Pentixafor uptake analysis.

### *In vivo* CXCR4 imaging of mice

Positron emission tomography scans of xenotransplanted MM.1S and OPM-2 tumors in NOD.CB17-*Prkdc*$^{scid}$/NCrHsd mice were performed as previously described (Graf *et al*, 2014). Briefly, mice were intravenously injected with 2.5 MBq/mouse [68Ga]Pentixafor or 9 MBq/mouse [18F]FDG and static images were acquired for 15 min starting 1 h post-injection on a μPET system (Inveon, Siemens, Erlangen, Germany). All mice received a [68Ga]Pentixafor PET scan and a second one with [18F]FDG the following day. Tumor-to-background ratios of tracer intensity were calculated by placing three-dimensional regions of interest within the tumors and in healthy tissue (background). In competition imaging assays, AMD3100 (2 mg/kg body weight, Selleck Chemicals) was intravenously injected immediately before [68Ga]Pentixafor was injected. PET scans were acquired 1 h post-PET tracer injection.

### PET/CT and PET/MR imaging studies

All [18F]FDG scans and 12/14 [68Ga]Pentixafor scans were performed on dedicated PET/CT scanners (Siemens Biograph mCT 64; Siemens Medical Solutions, Germany), whereas 2 of 14 [68Ga] Pentixafor scans were performed on a PET/MRI device (Siemens Biograph mMR; Siemens Medical Solutions, Germany). Before

                    

acquisition of [18F]FDG PET scans, patients fasted for at least 6 h prior to injection of a standard dose of 4.5 MBq per kg body weight. [18F]FDG was only injected if blood glucose levels were < 180 mg/dl. Prior to [68Ga]Pentixafor scans, patients fasted for at least 4 h. Injected activities ranged from 90 to 205 MBq. Corresponding CT low-dose scans for attenuation correction were acquired using a low-dose protocol (20 mAs, 120 keV, a $512 \times 512$ matrix, 5 mm slice thickness, increment of 30 mm/s, rotation time of 0.5 s, and pitch index of 0.8) including the base of the skull to the proximal thighs. In PET/MR, first a coronal 2-point Dixon 3D volumetric interpolated examination (VIBE) T1 weighted (T1w) MR sequence was performed for generation of attenuation maps as recently published (Drzezga et al, 2012). In addition, both a coronal T1 TSE (TR/TE 600/8.7, slice thickness 5 mm, matrix $384 \times 230$) and a T2w STIR (short τ inversion recovery) sequence with fat suppression (TR/TE/TI 5,000 ms/56 ms/220 ms, slice thickness 5 mm, matrix $106 \times 256$) were acquired. Consecutively, PET emission data were acquired in three-dimensional mode with a $200 \times 200$ matrix with 2–3 min emission time per bed position. After decay and scatter correction, PET data were reconstructed iteratively with attenuation correction using a dedicated software (Siemens Esoft).

## PET, CT and MR analysis

All CT and MRI scans were scored by a board-certified radiologist, and all PET scans were scored by a board-certified nuclear medicine physician. All PET scans were interpreted in a binary visual fashion as positive for disease or negative for disease according to the criteria previously described (Zamagni et al, 2011). Briefly, the presence of focal areas of detectable increased tracer uptake within bones (e.g., more intense than background bone marrow (BM) uptake excluding articular processes, with or without any underlying lesion identified by CT) were rated positive. If the scan was rated as positive, disease manifestations were rated as either intra- and/or extramedullary. Intramedullary disease was then separately assessed for 13 regions including head, spine (cervical, thoracic, lumbar), sacrum, pelvis (left and right), upper (left and right) and lower (left and right) extremities, as well as rib cage (left and right). Involvement of the sternum went along in all patients with rib cage manifestations and was therefore not separately assessed. Semi-quantitative analysis comprised calculation of maximum standardized uptake values ($SUV_{max}$) as well as $SUV_{mean}$ by 2D ROIs with a diameter of 1.5 cm around the hottest pixel. Up to 3 lesions were recorded, if subjects presented with more than 3 focal lesions (FL); they were categorized into the subgroup > 3 FL. Lesions in the appendicular skeleton were divided from those in the axial portions. Diffuse BM involvement was considered if the tracer uptake was diffusely increased with a $SUV_{max}$ equal to, or greater than, the uptake in the spleen. The presence of extramedullary disease (EMD), defined as [18F]FDG-avid tissue that, according to CT examination, was not contiguous to bone and arose in soft tissue, was described by location and number of lesion. Paramedullary disease arising from bone was considered as a lesion but not as EMD.

CT and MR scans were read as outlined previously (Angtuaco et al, 2004). In CT, any osteolytic changes not related to degenerative or other benign changes (e.g., hemangioma) were rated as suspicious for MM. In MR, any focal lesion presenting with low signal intensity on T1w TSE images (signal intensity not higher than

## The paper explained

### Problem

Malignancies of the hematopoietic or lymphoid tissues are mostly considered systemic diseases involving the whole body. Therefore, systemic treatment approaches are applied, for example, classical chemotherapy or novel drugs. Cancer cells evade such potentially effective and curative treatment by localizing to a putative protective niche, from where relapse is thought to occur. The chemokine/chemokine receptor axis SDF-1alpha/CXCR4 is a major determinant for recruiting cancer cells to this protective niche.

### Results

Here, we provide the first evaluation of in vivo CXCR4 imaging in a series of patients with a particular cancer of the lymphoid system, multiple myeloma, using the CXCR4-specific PET tracer Pentixafor in comparison with the clinically established tracer FDG. We identify myeloma manifestations that are positive for CXCR4 uptake and establish the use of an in vivo biomarker imaging technique for future therapeutic/theranostic purposes.

### Impact

Our findings identify the CXCR4 PET tracer Pentixafor as a novel tool for in vivo imaging of multiple myeloma. This tracer is suitable for identifying patients who could be treated with CXCR4-directed drugs, thus reflecting an in vivo biomarker. Labeling of Pentixafor or derived peptides with radionuclides or drug conjugation seems suitable for therapeutic targeting of the cancer cell and its protecting niche.

surrounding muscle) and high signal intensity on T2-weighted sequences and STIR images were judged as suspicious for MM. For intramedullary disease, the same regions were rated as in PET. In addition, any extramedullary lesions defined by extramedullary soft-tissue formations were noted.

## Immunohistochemistry and immunofluorescence of patient biopsy material

For immunohistochemistry, the following antibodies were used: anti-CD138 mouse (B-A38) monoclonal antibody (Cell Marque, CA, USA) and anti-CXCR4 rabbit polyclonal antibody (Abcam). After deparaffinization and rehydration, the slides were placed in a pressure cooker in 0.01 M citrate buffer (pH 6.0) and were heated for 7 min. Incubation with the different antibodies was carried out overnight at 4°C. Detection was performed with DAKO en vision system according to the manufacturer's protocol. For double immunofluorescence, primary antibodies were detected by incubation with the following secondary antibodies: donkey anti-rabbit conjugated with Dylight 488 (Jackson ImmunoResearch, Suffolk, UK) and donkey anti-mouse conjugated with Cy5 (Jackson Immuno-Research). After incubation of slides with conjugated secondary antibody for 30 min, slides were counterstained and mounted with mounting medium (Vectashield, Vector laboratories, Burlingame, CA, USA).

## Statistical analysis

All statistical tests were performed using SPSS Statistics version 22 (IBM) or GraphPad Prism (GraphPad Software). $P$-values < 0.05

were considered statistically significant. Quantitative values were expressed as mean ± standard deviation or standard error of the mean (SEM) or standard deviation (SD) as indicated. Comparisons of related metric measurements were performed using Wilcoxon signed-rank test, and the Mann–Whitney *U*-test or Student's *t*-test was used to compare quantitative data between two independent samples. Analysis of variance (ANOVA) statistical test was used to analyze the differences between group means.

**Supplementary information** for this article is available online: http://embomolmed.embopress.org

## Acknowledgements

We are grateful to E. Schafnitzel, K. Vollmer, N. Wildegger, and V. Hollnburger for assistance with BM sample flow cytometry. We thank the Hematology/ Medical Oncology and Nuclear Medicine staff members at the institutions in Munich and Würzburg for their support. HJW, UK, and MS are supported by the Deutsche Forschungsgemeinschaft (DFG, SFB824). UK was further supported by Deutsche Krebshilfe (grant 111305) and DFG (grant KE 222/7-1). This work received support from the German Cancer Consortium (DKTK).

## Author contributions

KPA, MS, EP, KL, KF, MR, AR, AS, KS, SK, KG, HS, SH, EH, and SS performed and interpreted the experiments. ME, CG, AKB, AJB, EH, CL, and KH analyzed and interpreted the imaging data. SK and HE contributed retrospective patient data. UK, KH, AKB, CP, MS, KL, and HJW designed the study, interpreted the data, and wrote the manuscript. All authors critically reviewed and approved the final manuscript.

## Conflict of interest

HJ Wester and S Kropf are CEOs of Scintomics, the distributor of Pentixafor. The remaining authors declare that they have no conflict of interest.

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
