## [Review Process File · EMBO Molecular Medicine]

In vivo molecular imaging of chemokine receptor CXCR4 expression in patients with advanced multiple myeloma

Kathrin Philipp-Abbrederis, Ken Herrmann, Stefan Knop, Margret Schottelius, Matthias Eiber, Katharina Lückerath, Elke Pietschmann, Stefan Habringer, Carlos Gerngrofl, Katharina Franke, Martina Rudelius, Andreas Schirbel, Constantin Lapa, Kristina Schwamborn, Sabine Steidle, Elena Hartmann, Andreas Rosenwald, Saskia Kropf, Ambros J. Beer, Christian Peschel, Hermann Einsele, Andreas K. Buck, Markus Schwaiger, Katharina Götze, Hans-Jürgen Wester, and Ulrich Keller

Corresponding author: Ulrich Keller, Technische Universität München

Review timeline:

Submission date:	01 October 2014
Editorial Decision:	04 November 2014
Revision received:	23 December 2014
Editorial Decision:	19 January 2015
Revision received:	26 January 2015
Accepted:	26 January 2015

Transaction Report:

Editor: Roberto Buccione

1st Editorial Decision

04 November 2014

Thank you for the submission of your manuscript to EMBO Molecular Medicine. We have now heard back from the three Reviewers whom we asked to evaluate your manuscript.

You will see that the Reviewers are generally supportive of your work although all three and Reviewer 2 in particular, express significant and overlapping concerns that prevent us from considering publication at this time.

Reviewer 1, in addition to requesting a number of important clarification/amendments in your manuscript, would like to see data on the correlation between the PET signal and CXCR4 expression and additional comparative analysis from the patients with extensive bone disease.

Reviewer 2 is more reserved and points to a number of experimental issues that require your action. S/he is especially concerned that the specificity of detection of CXCR4 is based on correlative evidence and suggests LOF studies to test this conclusion more thoroughly. Similarly, s/he would like you to include CXCR4-negative MM patients in the study. Clearly these concerns in part overlap with those of Reviewer 1. Reviewer 2 also lists other issues that require your action.

Reviewer 3 also mentions the specificity/sensitivity issues due also to the lack of appropriate controls as mentioned by the reviewer 2.

Considering all the above, while publication of the paper cannot be considered at this stage, we

would be pleased to consider revised submission, with the understanding that the Reviewers' concerns must be fully addressed, with additional experimental data where appropriate and that acceptance of the manuscript will entail a second round of review.

We would like to remind you that it is EMBO Molecular Medicine policy to allow a single round of revision only and that, therefore, acceptance or rejection of the manuscript will depend on the completeness of your responses included in the next, final version of the manuscript.

As you know, EMBO Molecular Medicine has a "scooping protection" policy, whereby similar findings that are published by others during review or revision are not a criterion for rejection. However, I do ask you to get in touch with us after three months if you have not completed your revision, to update us on the status. Please also contact us as soon as possible if similar work is published elsewhere.

Please note that EMBO Molecular Medicine now requires a complete author checklist (<http://embomolmed.embopress.org/authorguide#editorial3>) to be submitted with all revised manuscripts. Provision of the author checklist is mandatory at revision stage; the checklist is designed to enhance and standardize reporting of key information in research papers and to support reanalysis and repetition of experiments by the community. The list covers key information for figure panels and captions and focuses on statistics, the reporting of reagents, animal models and human subject-derived data, as well as guidance to optimise data accessibility. I am attaching the checklist here for your convenience; Should you have problems opening the attachment, please download it using the above link.

I look forward to seeing a revised form of your manuscript as soon as possible.

***** Reviewer's comments *****

Referee #1 (Remarks):

This is a well written and very interesting manuscript that exemplifies translational research. Some issues should be clarified.

Why was there not visible uptake of the radiotracer in normal CXCR-4 expressing mouse organs? Please explain in one sentence (species related; expression level?).

Patient studies:

Do the authors have any data on correlation between the PET imaging signal and CXCR-4 expression levels in biopsy sites?

Please clarify if the 14 patients undergoing imaging studies were the same 14 patients mentioned in the section "Frequency of MM patients with..." If so, and if all 14 patients showed similarly high CXCR-4 expression in flow cytometry, why was the Ga-68 pentixafor scan negative in 4/14 patients?

The image examples and data plots convincingly show the high uptake of Ga-68 pentixafor in MM sites. Nevertheless, there was only one patient with negative FDG but positive pentixafor scan. With regard to individual disease sites, pentixafor scan showed more lesions, but also missed some lesions seen on FDG scan. Interestingly, both agents provided complementary information in some patients. The authors later (in the Discussion) clarify that they do not primarily consider Ga-68 pentixafor as a diagnostic imaging agent, but rather as a necessary guide and selection tool for a theranostic approach. It would be helpful to spell this out clearly in the Introduction and in the Abstract.

Comparison with FDG: FDG remains a suboptimal agent for defining the extent of MM, although it may correlate with disease activity and prognosis. FDG thus cannot really be considered as "gold standard" against which pentixafor should be tested. In this regard, it would be interesting to see a

comparison of the MR and pentixafor findings in the 2 patients imaged with combined PET/MR (page 7) who showed extensive bone disease. It would be helpful if the authors could provide these data.

Discussion:

The statement that "low dose computed tomography is usually sufficient" (page 9) deserves a reference and some clarification. The CT is an excellent tool to show lytic lesions (which may prompt initiation of treatment) but the extent of bone marrow disease is clearly much better defined by MRI.

Please rephrase the sentence (page 9) starting with "Previous efforts..." --- the clause "most likely due to in site blood pool..." is unclear as written.

Methods:

Why did the injected activity for Ga-68 pentixafor vary so widely (90-205 MBq)? Was it based on body weight?

Minor issues:

Please be consistent in using past tense throughout the manuscript (e.g., abstract line 5: "provides" >> "provided").

Page 4: "Wester, in preparation" is not an acceptable reference. Please provide at least the reference for the abstract or some other detail.

Referee #2 (Remarks):

In this manuscript, Philipp-Abbrederis et al. report the use of [68Ga]Pentixafor-PET to identify multiple myeloma aggregates expressing surface CXCR4 in patients. The ultimate goal is to use this tool to more accurately identify patients with CXCR4-positive cancers, and to adapt this technique to deliver toxins to CXCR4-positive cancer cells.

Figure 1: CXCR4 expression in MM cells from BM.

Here the authors measure CXCR4 expression on MM and untransformed cells from patients.

1. The gating strategy to identify MM and untransformed subsets, shown in E1, is unclear (an unconventional way to show the data, and there is no text to explain). This should be explicit in the manuscript rather than "available upon request."
2. The authors do not detect CXCR4 on normal B cells or CD34+ progenitors, contrary to published results showing surface CXCR4 expression on these cells (e.g. Honczarenko et al. Blood 1999; Aiuti et al. Blood 1999). The authors should mention and explain the discrepancy. (Are they similarly missing CXCR4 on the ~50% of patients they describe as having CXCR4-negative MM?)
3. The legend to Figure 1C doesn't seem to describe the histogram-is it an intensity or a ratio, and if a ratio how is the value ~0 for myeloma cells:T cells?

Figure 2: [68Ga]Pentixafor-PET imaging of MM xenografts.

Here the authors offer initial evidence that their probe can detect MM in a xenograft model.

They suggest that the probe is specific based on a comparison of detection of 2 MM cell lines with different CXCR4 expression -- the line with higher CXCR4 expression, OPM-2, is detected more clearly than the line with lower CXCR4 expression, MM1-S. But this is simply a correlation, and OPM-2 is also better detected by FDG. To start to test specificity, the authors should knock down CXCR4 in the two lines and ask if this affects probe binding.

They argue in the results section that one important finding of the xenograft model is that the probe is not taken up in normal tissues with high CXCR4 expression. Yet in the discussion they state that the probe is selective for human CXCR4, so this is very misleading.

Figures 3, 4: Example PET scans.

Here the authors show that [68Ga]Pentixafor-PET detects MM lesions in human patients.

The specificity would have been more convincing had they included patients in their study with CXCR4-negative MM - almost half the initial cohort. And as the authors are proposing this as a vehicle to deliver toxins to tumors in vivo, they should have more rigorously assessed binding to normal cells - their conclusion that binding is negligible seems to be based on a qualitative look at the scans, and is surprising based on the known expression pattern of CXCR4.

Referee #3 (Comments on Novelty/Model System):

The authors convincingly demonstrates that the [68Ga] Pentixafor probe detects CXCR4. Given importance of CXCL12/CXCR4 axis in many tumors, the probe represents a valuable tool for in vivo detection of CXCR4.

Referee #3 (Remarks):

Philipp-Abbrederis et al assessed [68Ga] Pentixafor, a novel high affinity CXCR4- targeted probe for Positron Emission Tomography imaging, in both multiple myeloma (MM) xenograft models and advanced MM patients.

The authors used flow cytometry to assess levels of CXCR4 in MM cells derived from bone marrow biopsies from 25 patients. They found that in about half of the samples, the levels of CXCR4 were significantly upregulated in MM cells as compared to other hematopoietic populations. Next, the group tested whether [68Ga] Pentixafor was suitable for detecting CXCR4+ Multiple Myeloma cells in vivo by using a xenograft model of CXCR4-expressing cell lines and they found that the probe efficiently detected CXCR4-positive MM xenografts.

Overall the study convincingly demonstrates that the [68Ga] Pentixafor probe detects CXCR4 in MM cells, but the ability of the probe to distinguish between high and low levels of CXCR4, similarly to flow cytometry, needs to be supported by better controls.

1. In their in vivo xenograft studies the authors did not use an appropriate specificity control-- xenografted multiple myeloma tumor expressing low surface levels of CXCR4 (or at the very least xenografted 293T cells).
2. The authors compare PET/CT scans of multiple myeloma patients using either CXCR4-specific [68Ga] Pentixafor or a glucose analog [18F]FDG. It is not clear from the manuscript under which criteria the authors selected the patients depicted in the figures. Given that the authors performed a thorough analysis of CXCR4 expression in the bone marrow biopsies, it would be appropriate to compare [68Ga] Pentixafor PET/CT scans between groups of patients with MM cells expressing low and high levels of CXCR4, as detected by flow cytometry.

1st Revision - authors' response

23 December 2014

Referee #1 (Remarks):

“This is a well written and very interesting manuscript that exemplifies translational research. Some issues should be clarified.”

We thank Referee #1 for the positive evaluation of our body of work, and for the critical comments and suggestions, which we have addressed wherever feasible to improve the data presented in the manuscript.

Specific comments addressed:

#1. *“Why was there not visible uptake of the radiotracer in normal CXCR-4 expressing mouse organs? Please explain in one sentence (species related; expression level?).”*

Pentixafor (CPCR4-2) is a peptide that was developed to bind with high affinity to the human CXCR4 chemokine receptor but not mouse CXCR4 (Gourni et al, 2011). This is now more clearly stated within the manuscript.

With regard to the issue of CXCR4 expression level in normal human tissue and [⁶⁸Ga]Pentixafor-PET imaging we would like to refer to our response to comment #2 raised by Referee #2.

#2. *“Do the authors have any data on correlation between the PET imaging signal and CXCR-4 expression levels in biopsy sites?”*

The biopsy of CXCR4-positive sites identified by [⁶⁸Ga]Pentixafor-PET was not planned at any point during the imaging examinations. We are however currently preparing a study protocol that involves the pre-surgical imaging using the Pentixafor probe. These studies will allow directly correlating CXCR4 imaging by [⁶⁸Ga]Pentixafor-PET and PET parameters such as the mean/max standard uptake value (SUV) with CXCR4 surface expression of tumor cells assessed by immunofluorescence and immunohistochemistry, or in the case of imaging-directed bone marrow biopsies by CD184 flow cytometry. [Unpublished data omitted upon Author request].

#3. *“Please clarify if the 14 patients undergoing imaging studies were the same 14 patients mentioned in the section “Frequency of MM patients with...” If so, and if all 14 patients showed similarly high CXCR-4 expression in flow cytometry, why was the Ga-68 pentixafor scan negative in 4/14 patients?”*

The n=14 patients undergoing the imaging studies were not identical with the cohort of n=25 patients tested for CXCR4 positivity by flow cytometry. This is now clearly stated within the manuscript.

Flow cytometric analysis of the n=25 samples revealed positivity in 56% (n=14) of the samples tested. [⁶⁸Ga]Pentixafor-PET imaging showed visual positivity in 10 of 14 patients (71%). Thus, there seems to be a comparable proportion of positive patient subgroups within the two sample cohorts. At this point we however cannot and do not want to claim that there would be a complete concurrence of the data that could be obtained when assessing a PET-directed biopsy of myeloma foci by immunohistochemistry and/or immunofluorescence and/or flow cytometry. Again, the clinical examination was not designed to confirm CXCR4 positivity by biopsy of [⁶⁸Ga]Pentixafor-PET-positive sites. Such an analysis is also not realistic in patients with multiple lesions, where there might or even is supposed to be heterogeneity with regard to CXCR4 expression on myeloma sites even within each single patient.

#4. *“The image examples and data plots convincingly show the high uptake of Ga-68 pentixafor in MM sites. Nevertheless, there was only one patient with negative FDG but positive pentixafor scan. With regard to individual disease sites, pentixafor scan showed more lesions, but also missed some lesions seen on FDG scan. Interestingly, both agents provided complementary information in some patients. The authors later (in the Discussion) clarify that they do not primarily consider Ga-68 pentixafor as a diagnostic imaging agent, but rather as a necessary guide and selection tool for a theranostic approach. It would be helpful to spell this out clearly in the Introduction and in the Abstract.”*

We thank the Referee for this very important remark. We have therefore now placed corresponding text passages within the abstract as well as the introduction section of the manuscript. We fully

agree with the Referee that both PET imaging modalities could be complementary and that the Pentixafor probe is a highly promising tool for a theranostic treatment approach.

#5. *“Comparison with FDG: FDG remains a suboptimal agent for defining the extent of MM, although it may correlate with disease activity and prognosis. FDG thus cannot really be considered as “gold standard” against which pentixafor should be tested. In this regard, it would be interesting to see a comparison of the MR and pentixafor findings in the 2 patients imaged with combined PET/MR (page 7) who showed extensive bone disease. It would be helpful if the authors could provide these data.”*

We fully agree with the Referee that FDG-PET is not the gold standard imaging method for MM.

We have picked up the suggestion of Referee #2 and have performed the suggested analyses of the patients undergoing the PET-MRI scans. Select data from this analysis are now included into the manuscript as novel Figure 5. These images reveal the sensitive detection of myeloma foci visible by MRI and by [⁶⁸Ga]Pentixafor-PET. Former Figure 4 of the main manuscript is now placed into the Expanded View section (novel Figure E5). Since only 2 patients underwent the PET/MRI assessment a systematic comparison to CT for detected MM foci detection was not performed.

#6. *“The statement that “low dose computed tomography is usually sufficient” (page 9) deserves a reference and some clarification. The CT is an excellent tool to show lytic lesions (which may prompt initiation of treatment) but the extent of bone marrow disease is clearly much better defined by MRI.”*

We thank the Reviewer for this important comment and have accordingly clarified this in the manuscript text (discussion section, page 9) and supported this information by suitable references. Please also see our response to comment #5 above and novel Figure 5 in the manuscript.

#7. *“Please rephrase the sentence (page 9) starting with “Previous efforts...” --- the clause “most likely due to in site blood pool...” is unclear as written.”*

We thank the Referee for pointing to this statement and have accordingly removed the unclear phrasing.

#8. *“Why did the injected activity for Ga-68 pentixafor vary so widely (90-205 MBq)? Was it based on body weight?”*

The routine synthesis of [⁶⁸Ga]-Pentixafor was only recently established at the two participating institutions. As we had only limited experience we injected around 200 MBq in the first three patients (198/205/200 MBq). As we realized that [⁶⁸Ga] Pentixafor allows imaging with a low background uptake we decided to reduce the injected activity to 120-180 MBq depending on the body weight. Some lower injected activities were due to lower than expected yields during the synthesis.

#9. *“Please be consistent in using past tense throughout the manuscript (e.g., abstract line 5: “provides” >> “provided”).”*

We have thoroughly revised the manuscript according to the Referee’s remark.

#10. "Page 4: "Wester, in preparation" is not an acceptable reference. Please provide at least the reference for the abstract or some other detail."

Regarding the citation of unpublished data we had followed the style suggestion requested by EMBO Mol Med. The journal does not suggest using other referencing. For information of the Referee we are however happy to provide the requested information that is available from Web of Science:

Wester H, Keller U, Beer AB, Schottelius M, Hoffmann F, Kessler H, Schwaiger M (2013) [Ga-68]CPCR4.2-PET for Imaging of CXCR4-Chemokine Receptors opens a new and exciting field of clinical research. *Eur J Nucl Med Mol I* **40**: S152-S152 (Congress report).

Referee #2 (Remarks):

In this manuscript, Philipp-Abbrederis et al. report the use of [68Ga]Pentixafor-PET to identify multiple myeloma aggregates expressing surface CXCR4 in patients. The ultimate goal is to use this tool to more accurately identify patients with CXCR4-positive cancers, and to adapt this technique to deliver toxins to CXCR4-positive cancer cells.

Specific comments addressed:

#1. "Figure 1: CXCR4 expression in MM cells from BM. Here the authors measure CXCR4 expression on MM and untransformed cells from patients.

The gating strategy to identify MM and untransformed subsets, shown in E1, is unclear (an unconventional way to show the data, and there is no text to explain). This should be explicit in the manuscript rather than "available upon request."

We thank the Reviewer for pointing to this shortcoming in the initial version of the manuscript. We now provide a schematic approach of the gating strategy within novel Figure E1 including additional explanatory text. Novel Figure E2 shows the assessment of CXCR4 expression in an exemplary patient sample. The gating strategy used to assess plasma cell infiltration in the bone marrow is based on the classic CD45 vs SSC and CD45 vs CD38 gating strategy and has been established in our department using software and expertise from the manufacturer of the hardware (Beckman Coulter). Using this strategy we have successfully participated in national inter-laboratory comparison tests for quality assurance in hematology diagnostic centers (INSTAND Round Robin Tests) for >10 years. We however agree that this method is not comprehensive for the newly (at least in clinical trials) established detection of minimal residual disease.

#2. "The authors do not detect CXCR4 on normal B cells or CD34+ progenitors, contrary to published results showing surface CXCR4 expression on these cells (e.g. Honczarenko et al. *Blood* 1999; Aiuti et al. *Blood* 1999). The authors should mention and explain the discrepancy. (Are they similarly missing CXCR4 on the ~50% of patients they describe as having CXCR4-negative MM?)"

We used a flow cytometry method that only allows relative assessment of CXCR4 expression, not quantification of cell surface receptor levels, which would require adding beads. This is now clearly stated within the methods section of the manuscript. To our knowledge there is no reference method for such assessment of CXCR4 expression available, and using an isotype control antibody represents a frequently used and widely accepted method, which we therefore applied. Importantly, our data show that in a substantial fraction of patients with MM plasma cells express CXCR4 at very high levels compared to intraindividual populations of the tested bone marrow samples. We now

clearly state this within the manuscript. Importantly, our [⁶⁸Ga]-Pentixafor-PET scans reveal that normal lymph nodes and the bone marrow, despite harbouring the mentioned normal cell subpopulations, are not characterized by [⁶⁸Ga]-Pentixafor binding, confirming that high expression of CXCR4 can be detected using CXCR4-directed PET.

Data from several previously published manuscripts have shown that cancer cells are often characterized by high CXCR4 expression (Teicher & Fricker, 2010; Weilbaecher et al, 2011) and our data support these findings. Furthermore, several of these manuscripts acknowledge that there is substantial heterogeneity with regard to CXCR4 expression in various malignancies. Thus, there is no contradiction in the data shown in Figure 1 and we fully acknowledge that lymphocytes and CD34+ cells express functional CXCR4, which is however fairly low as compared to the MM cells that stain positive in our analyses. This is now also clearly stated within the manuscript.[Unpublished data omitted upon Author request].

In conclusion, our flow cytometry, immunohistochemical and imaging data reveal that a substantial proportion of patients with MM are characterized by high tumor CXCR4 expression.

#3. “The legend to Figure 1C doesn't seem to describe the histogram-is it an intensity or a ratio, and if a ratio how is the value ~0 for myeloma cells:T cells?”

We apologize that the legend to Figure 1c was misleading and we agree with the reviewer that the legend as shown needed to be improved for better understanding. Accordingly, we have revised the legend of Fig. 1, the supplemental Figures E1 and E2, and the methods section. Briefly, the data shown in Fig. 1c represents the median fluorescence intensity of CD184 staining relative to isotype control for each individual patient. Since staining of CD34+ cells and T cells was weak, values may be close to 0 relative to isotype control.

#4. “Figure 2: [⁶⁸Ga]Pentixafor-PET imaging of MM xenografts. Here the authors offer initial evidence that their probe can detect MM in a xenograft model. They suggest that the probe is specific based on a comparison of detection of 2 MM cell lines with different CXCR4 expression -- the line with higher CXCR4 expression, OPM-2, is detected more clearly than the line with lower CXCR4 expression, MM1-S. But this is simply a correlation, and OPM-2 is also better detected by FDG. To start to test specificity, the authors should knock down CXCR4 in the two lines and ask if this affects probe binding.”

Only stable RNA interference by either retroviral or lentiviral transduction followed by selection using a suitable marker (usually GFP) would allow efficient knock-down in MM cell lines. We have performed such efforts within our previous work without success (e.g. Dechow, Steidle et al., J Clin Invest. 2014, 124:5263-74). We have therefore chosen an alternative suitable approach by treating MM cell lines or MM xenografted mice with a Pentixafor competitor, AMD3100 (Plerixafor) that binds to the same epitope. The data provided in novel Figure 2H and supplemental Figure E3 provide clear evidence that the binding of the novel PET probe Pentixafor is efficiently blocked by competing access AMD3100/Plerixafor, strongly suggesting specificity. In addition, the [⁶⁸Ga]-Pentixafor-PET patient imaging data revealed a very low level of background activity (see Figure 3 and in particular novel Figure 5), pointing to selective binding of Pentixafor to MM foci in vivo.

#5. “They argue in the results section that one important finding of the xenograft model is that the probe is not taken up in normal tissues with high CXCR4 expression. Yet in the discussion they state that the probe is selective for human CXCR4, so this is very misleading.”

We thank the Reviewer for pointing to this imprecise statement. We have previously shown that CPC4-2 (Pentixafor) is a human CXCR4-specific probe (Gourni et al, 2011). In order to not mislead the readership we clarified that we did not expect binding of the CXCR4 probe Pentixafor to mouse tissue. Accordingly, and since Pentixafor is secreted renally, only the renal pelvis and the bladder show positivity in the [⁶⁸Ga]Pentixafor-PET scans next to MM lesions.

#6. “Figures 3, 4: Example PET scans. Here the authors show that [68Ga]Pentixafor-PET detects MM lesions in human patients. The specificity would have been more convincing had they included patients in their study with CXCR4-negative MM - almost half the initial cohort.”

Please see our response to comment #3 of Referee #1. Briefly, of the n=14 MM patients assessed by Pentixafor-PET imaging, n=10 patients were found positive. Thus, there were also patients that had negative imaging for CXCR4. In order to clarify this issue for the Reviewer and also the reader, we have now included a graphic illustration of positivity/negativity for CXCR4 assessed by PET imaging as well as for FDG-PET (novel Figure 4). We have also included images of a MM patient negative for CXCR4 imaging as suggested by this Referee (novel Expanded View Figure E4). For presentation of a patient with RCC that was proven negative for CXCR4 expression upon surgical removal and histologic assessment please see Figure 1 (Referees only).

#7. “And as the authors are proposing this as a vehicle to deliver toxins to tumors in vivo, they should have more rigorously assessed binding to normal cells - their conclusion that binding is negligible seems to be based on a qualitative look at the scans, and is surprising based on the known expression pattern of CXCR4.”

We fully agree with the Reviewer that there is a real possibility and likelihood that using a radionuclide labelled-CXCR4 binding peptide also causes side effects by binding to normal lymphocytes or CD34+ hematopoietic stem/progenitor cells. This is now stated within the discussion. A corresponding trial using a Pentixafor-derived peptide that can be easily labelled with [¹⁷⁷Lu] has been submitted to the Deutsche Krebshilfe (German Cancer Aid) for funding the further clinical development. For this proof-of-concept study it would be however too early to include details on such a future clearly tangible approach.

Referee #3 (Comments on Novelty/Model System):

“The authors convincingly demonstrates that the [68Ga] Pentixafor probe detects CXCR4. Given importance of CXCL12/CXCR4 axis in many tumors, the probe represents a valuable tool for in vivo detection of CXCR4.”

We thank Referee #3 for the positive evaluation of our work.

Referee #3 (Remarks):

Philipp-Abbrederis et al assessed [68Ga] Pentixafor, a novel high affinity CXCR4- targeted probe for Positron Emission Tomography imaging, in both multiple myeloma (MM) xenograft models and advanced MM patients.

The authors used flow cytometry to assess levels of CXCR4 in MM cells derived from bone marrow biopsies from 25 patients. They found that in about half of the samples, the levels of CXCR4 were significantly upregulated in MM cells as compared to other hematopoietic populations. Next, the group tested whether [68Ga] Pentixafor was suitable for detecting CXCR4+ Multiple Myeloma cells in vivo by using a xenograft model of CXCR4-expressing cell lines and they found that the probe efficiently detected CXCR4-positive MM xenografts.

Specific comments addressed:

#1. Overall the study convincingly demonstrates that the [68Ga] Pentixafor probe detects CXCR4 in MM cells, but the ability of the probe to distinguish between high and low levels of CXCR4, similarly to flow cytometry, needs to be supported by better controls.

Please see our response to comment #2 and #7 of Reviewer #2. We however feel that the in vivo patient imaging data shown within this manuscript also strongly indicate that binding to low level CXCR4-expression lymphocytes and CD34+ hematopoietic stem and progenitor cells (HSPC) does not result in Pentixafor-PET positivity, in particular of the bone marrow where HSPC reside and the lymph nodes where B- and T-cells reside. We also would like to point to novel Figure 2 (Referees only) where we analyzed a patient with lymph node infiltration of MM and clearly find evidence that malignant cells may express very high levels of CXCR4 as compared to lymphocytes that reside within the follicle B-zone or the T-zone of the lymph node.

#2. "1. In their in vivo xenograft studies the authors did not use an appropriate specificity control--xenografted multiple myeloma tumor expressing low surface levels of CXCR4 (or at the very least xenografted 293T cells)."

Please see our response to comment #4 of Referee #2.

#3. "The authors compare PET/CT scans of multiple myeloma patients using either CXCR4-specific [68Ga] Pentixafor or a glucose analog [18F]FDG. It is not clear from the manuscript under which criteria the authors selected the patients depicted in the figures. Given that the authors performed a thorough analysis of CXCR4 expression in the bone marrow biopsies, it would be appropriate to compare [68Ga] Pentixafor PET/CT scans between groups of patients with MM cells expressing low and high levels of CXCR4, as detected by flow cytometry."

As already pointed out in our response to comments #1 of Referee #1 and comments #6 of Reviewer #2 the imaging protocol did not include a biopsy performed for correlation purposes. Also, we have clarified that the n=25 patients that had routine bone marrow biopsies for staging are not identical with the group of n=14 patients undergoing the imaging study. A study where all patients and selected tumor sites will be analyzed by such means is currently submitted to the German agency for radiation safety (BfS) and to the ethics committee for approval. We furthermore would like to refer to novel Figure #1 for Reviewers only, where a patient suffering from papillary renal cell cancer underwent CXCR4 imaging and the thereafter resected tumor was shown negative for CXCR4 expression by immunohistochemistry.

Additional reference:

Gourni E, Demmer O, Schottelius M, D'Alessandria C, Schulz S, Dijkgraaf I, Schumacher U, Schwaiger M, Kessler H, Wester HJ (2011) PET of CXCR4 expression by a (68)Ga-labeled highly specific targeted contrast agent. *Journal of nuclear medicine : official publication, Society of Nuclear Medicine* **52**: 1803-1810.

Thank you for the submission of your revised manuscript to EMBO Molecular Medicine. We have now received the enclosed reports from the referees that were asked to re-assess it. As you will see the reviewers are now globally supportive and I am pleased to inform you that we will be able to accept your manuscript pending the following final amendments:

1) As per our Author Guidelines, the description of all reported data (including in the supplementary information) that includes statistical testing must state the name of the statistical test used to generate error bars and P values, the number (n) of independent experiments underlying each data point (not replicate measures of one sample), and the actual P value for each test (not merely 'significant' or ' $P < 0.05$ ').

2) Please note that all manuscript figure files must be uploaded as individual files.

3) We are now encouraging the publication of source data, particularly, but not limited to, electrophoretic gels and blots, with the aim of making primary data more accessible and transparent to the reader. Would you be willing to provide a PDF file the original, uncropped and unprocessed scans of the gel used in the manuscript? The PDF files should be labeled with the appropriate figure/panel number, and should have molecular weight markers; further annotation may be useful but is not essential. Any other additional source data would be welcome. This information will be published online with the article as supplementary "Source Data" files. If you have any questions regarding this just contact me.

4) Every published paper now includes a 'Synopsis' to further enhance discoverability (not to be confused with the "The Paper Explained" section). Synopses are displayed on the journal webpage and are freely accessible to all readers. They include a short standfirst (to be written by the editor) as well as 2-5 one sentence bullet points that summarise the paper (to be written by the author). Please provide the short list of bullet points that summarise the key NEW findings. The bullet points should be designed to be complementary to the abstract - i.e. not repeat the same text. We encourage inclusion of key acronyms and quantitative information. Please use the passive voice. Please attach these in a separate file or send them by email, we will incorporate them accordingly.

Please submit your revised manuscript within two weeks. I look forward to seeing a revised form of your manuscript as soon as possible.

***** Reviewer's comments *****

Referee #1 (Remarks):

The authors have addressed all questions and concerns satisfactorily. I feel that the manuscript is now acceptable for publication.

Referee #2 (Remarks):

The authors have addressed my concerns.

Referee #3 (Comments on Novelty/Model System):

All the experiment are now presented with appropriate controls. Because CXCL12/CXCR4 axis is important in many different cancers, the work is of high medical significance and will be of interest to readers of EMBO Molecular Medicine.

Referee #3 (Remarks):

All specific points were properly addressed and I now strongly recommend the manuscript for publication.